# Equipotent Dose and Cost Comparison of Atracurium and Rocuronium in Laboratory Pigs Anesthetized with Propofol

**DOI:** 10.3390/ani15131854

**Published:** 2025-06-23

**Authors:** Eleonora Benetti, Alessandro Mirra, Olivier Louis Levionnois

**Affiliations:** 1Anaesthesiology and Pain Therapy Section, Department of Clinical Veterinary Medicine, Vetsuisse Faculty, University of Bern, 3012 Bern, Switzerland; eleonora.benetti@unibe.ch (E.B.); alessandro.mirra@umontreal.ca (A.M.); 2Department of Clinical Sciences, Faculty of Veterinary Medicine, University of Montreal, Saint-Hyacinthe, QC J2S 8H5, Canada

**Keywords:** neuromuscular blocking agents (NMBAs), atracurium, rocuronium, cost-effectiveness, swine anesthesia, pharmacoeconomics

## Abstract

Neuromuscular blockers are often used to achieve muscle relaxation during surgery in laboratory pigs. However, the lack of specific dosing guidelines for this species often leads researchers to rely on data derived from other animal species, with the risk of inaccuracies. In this study, we compared two commonly used muscle relaxants, atracurium and rocuronium, in 12 healthy laboratory pigs anesthetized with propofol. The animals were divided into two groups of six, and each group received one of the two drugs. The dose of each drug was carefully titrated to maintain a stable level of muscle relaxation. We observed that although atracurium required a lower continuous dose than rocuronium, its cost was significantly higher. The starting dose of the two drugs was similar, but rocuronium proved to be less expensive in the Swiss context. Furthermore, the study showed that pigs require higher doses than other species, highlighting the need for dedicated guidelines.

## 1. Introduction

Neuromuscular blocking agents (NMBAs) play a crucial role in the framework of a comprehensive anesthetic plan. Non-depolarizing NMBAs function to induce temporary muscle paralysis by competitively inhibiting acetylcholine at nicotinic receptors, preventing normal neuromuscular transmission. Their addition at the induction of anesthesia has been reported to ease endotracheal intubation in both humans [1,2] and pigs [3,4]. Moreover, they are essential in achieving adequate muscle relaxation during specific operations (e.g., ophthalmic, laparotomy), allowing surgeons to perform procedures with greater precision and ease, significantly reducing the risk of muscle-related issues and improving surgical outcomes [5,6,7,8].

Particularly in the domain of biomedical research, the use of non-depolarizing NMBAs is common during surgical procedures in pigs. A review of research studies conducted between 2012 and 2014 revealed that NMBAs were administered in approximately 20% of the cases [6], a higher number than in veterinary practice for small domestic animals [9]. Unfortunately, there is limited information available on the species-specific dosing of NMBAs for pigs, and those reported are often extrapolated from other species.

To date, few studies have reported effective doses for atracurium and rocuronium in pigs. For experimental subjects, intravenous doses of atracurium ranging from 0.4 to 2.5 mg/kg [10,11,12] and rocuronium from 0.5 to 5 mg/kg have been documented [13,14,15,16]. However, no formal dosing guidelines have yet been established for their use in this species. Both drugs are classified as non-depolarizing NMBAs and are well-documented in porcine medical research [10,11,14,15,17]. While they exhibit comparable pharmacodynamic profiles, metabolic pathways differ and may influence the choice between them. Atracurium undergoes organ-independent metabolism via Hoffmann elimination, ensuring consistent clearance even in cases of renal or hepatic impairment or during prolonged administration. In contrast, rocuronium is primarily metabolized by the liver, with a minor portion excreted renally [11,13,17].

Despite these differences, pharmacokinetic considerations are rarely of practical relevance, and current literature does not favor the use of one agent over the other. Another consideration that may influence drug selection is the cost required to maintain comparable neuromuscular relaxation. This study aims to evaluate the cost-effectiveness of atracurium and rocuronium within the Swiss healthcare system in laboratory swine undergoing propofol anesthesia at equipotent infusion rates. The hypothesis to be tested is whether there is a relevant difference in cost between the two treatments that would lead to a preference for one drug over the other.

## 2. Materials and Methods

### 2.1. Ethical Statement

This study was conducted in accordance with the Committee for Animal Experiments of the Canton of Bern, Switzerland (Cantonal no. BE65/2023). Each pig underwent a single experimental trial, at the end of which euthanasia was performed (Pentobarbital, Euthasol 40%, Virbac AG, Opfikon, Switzerland; 150 mg/kg intravenous (IV)).

### 2.2. Experimental Design

Data collection presented here was performed concomitantly with another study objective on the evaluation of propofol-induced modulation of the electroencephalographic activity. Therefore, variations in the propofol administration rate by target-controlled infusion were performed during the study. The present investigation was designed as a prospective, blinded, randomized trial. Sample size analysis revealed that at least 11 pigs needed to be included to significantly detect a relevant difference in cost between the two groups (paired Wilcoxon signed rank test, α = 0.05, β = 0.8, effect size = 1, difference = 10%, standard deviation = 10%, GPower v3.1.9.7, Kiel, Germany).

### 2.3. Animals

Twelve pigs (phenotype Edelschwein) of both sexes (7 females and 5 males) with a body mass of 34.43 ± 8.17 kg (mean ± standard deviation (SD)) and of 12 ± 0.6 weeks of age were included in the study. Pigs were collected from the farm of origin between two and five days before the experiment, in groups of at least two animals, and brought to the animal facility of the University of Bern. Inclusion criteria were: healthy at physical examination (including normal feces), 9 to 14 weeks of age, and a body mass of at least 22 kg at inclusion. The animals were housed in single boxes. The animals were fed three times per day and had ad libitum access to water. Visual and auditory contact was allowed during the whole stay. In addition, environmental enrichments were positioned inside the boxes (e.g., straw, ropes). General physical examination (including rectal temperature) was performed at least once a day by a veterinarian. In the event of abnormal findings or any suspicion that the physical status score was higher than ASA-1 (according to the American Society of Anesthesiologists), the animal would be excluded and replaced.

### 2.4. Instrumentation

Instrumentation, anesthesia management and neuromuscular monitoring were performed by the same experienced anesthesiologists throughout the study (O.L.L., A.M.). On their experimental day, the animals were examined clinically and brought to the experimental room. They were allowed to acclimatize (between 30 and 60 min) and then placed into a sling. A local anesthetic cream (eutectic mixture of prilocaine and lidocaine, 5%, Anesderm, Pierre Fabre Dermo-Kosmetik GmbH, Freiburg, Germany) was applied over both ears and tail at least 45 min before placing an auricular venous catheter aseptically (Jelco 2 IV Catheter Radiopaque, 22 G × 25 mm, Smiths Medical ASD, Inc., Minneapolis, MN, USA). An isotonic solution (Ringer-Lactate Freeflex, 1000 mL, Fresenius-Kabi, Kriens, Switzerland) was administered during the whole anesthetic intravenously at a rate of 2 mL/kg/h (Infusomat Space pump, B.Braun, Sempach, Switzerland).

### 2.5. Anesthesia

Firstly, an intravenous (IV) infusion of propofol (Propofol 1% MCT, Fresenius Kabi, Kriens, Switzerland) was started without other premedication using a syringe pump (Orchestra, Base A + Module DPS visio, Fresenius Kabi, Kriens, Switzerland). Propofol administration rate was adjusted using target-controlled infusion (TCI, computer-controlled infusion pump software, CCIP v.3, Chinese Hong Kong University, Hong Kong, China), increasing progressively predicted propofol plasma concentration under tight facemask preoxygenation (4 L/min, 100% oxygen) until intubation was possible. Briefly, targeted propofol concentration was increased by steps of 3 μg/mL every 3 min by means of TCI until jaw relaxation was achieved, followed by steps of 1 μg/mL every minute until successful gentle endotracheal intubation. The endotracheal tube was connected to a semi-closed rebreathing system, and volume-controlled lung ventilation was started (FiO_2_ = 1, Aespire view, General Electrics Healthcare, Glattbrugg, Switzerland). Ventilatory parameters (10–12 mL/kg tidal volume, 6–10/min respiratory rate) were adjusted individually to maintain an end-tidal carbon dioxide partial pressure (ETCO_2_) of 40 mmHg, confirmed by arterial blood gas analysis, and adjusted accordingly.

After a short instrumentation period (electroencephalography) at a TCI propofol concentration maintaining a moderate depth of anesthesia (corresponding to approximately 12 μg/mL and 0.5 mg/kg/min), the individual predicted plasma concentration of propofol corresponding to signs of light depth of anesthesia (light palpebral reflex, no movements, reduced jaw tonus) was determined following an up-and-down titration method. This level was defined as Prop_Target_DoA_0% for each animal (corresponding to approximately 7.5 μg/mL and 0.3 mg/kg/min). During the experiment, following a standard protocol to investigate electroencephalographic modulation, the propofol infusion rate was then adjusted to target several levels between 1.25-fold (Prop_Target_DoA_25%) and 2.20-fold (Prop_Target_DoA_120%) of the individual Prop_Target_DoA_0%. Maintaining the propofol dose above Prop_Target_DoA_25% prevented the return of consciousness during NMBA administration.

At two occasions during general anesthesia, ETCO_2_ was increased to 80 mmHg for 15–20 min by adjusting ventilatory parameters according to the needs of the main aim of the study (NMBA data were excluded during this period, see below). Ventilation parameters, heart rate, respiratory rate, invasive blood pressure, ETCO_2_, oxygen saturation (SpO_2_), tidal volume, and positive inspiratory pressure (PIP) were continuously monitored (Carescape, General Electrics Healthcare, Glattbrugg, Switzerland) and representative values recorded every 5 min. Rectal body temperature was recorded approximately hourly.

### 2.6. Train-of-Four Measurements and NMBA Administration

The pigs were randomly assigned to receive either rocuronium (group R, Esmeron 10mg/mL, MSD Merck Sharp & Dohme AG, Luzern, Switzerland) or atracurium (group A, Atracurium Labatec 25 10mg/mL, Labatec Pharma SA, Meyrin, Switzerland). Group allocation was performed using a randomization application (randomizer.com, 2 groups, *n* = 12 in 2 blocks of 6) without distinction based on sex, body mass, or age. The veterinarians conducting the experiment remained blind to the treatment until the end of the study.

Syringes were prepared by another person not involved in the study and were all filled with the same volume. Based on the doses reported in the literature and the concentration of the injectable solutions used, the infusion rate of atracurium was expected to be approximately half of rocuronium’s rate and was therefore diluted 1:1 with physiological saline solution (Ringer-Acetat, Fresenius Kabi, Kriens, Switzerland). The animals were maintained in a prone position within a sling during the whole anesthesia.

To assess the level of neuromuscular blockage, a TOF-Watch SX (Organon Laboratories Limited, Cambridge, UK) was used. Two needles were inserted subcutaneously in the left front limb: one around 2 cm over the olecranon, the second one below it, at around 4 cm from the other one, over to the ulnar nerve, on the latero-palmar aspect of the front limb. The three-dimensional acceleration transducer was taped between the two digits, facing cranially. The limb was then slightly extended cranially with adhesive tape to prevent movements not directly related to the electrical stimulation, while the hoof was allowed to move freely. Before administering the NMBA, the Train-of-Four (TOF) ratio was measured, without prior calibration. Initial stimulation intensity was set at 50 mA.

If the TOF ratio was below 100% or the muscular response was deemed inadequate, the electrodes were repositioned. Conversely, if the muscular response was excessively strong, the stimulation intensity was gradually decreased (by steps of 5 mA) while keeping the TOF ratio at 100%. In the first step (Phase 1), the dose of NMBA required to achieve a TOF ratio of 0% (i.e., absence of the fourth twitch) was determined, followed in a second step by titration of the infusion rate to maintain a TOF count between 3 and 4 during the course of anesthesia (Phase 2). Initially, each pig received an IV bolus of NMBA at 0.2 mL/kg (corresponding to 2 mg/kg rocuronium or 1 mg/kg atracurium) administered over 1 min, immediately followed by a continuous infusion at 0.2 mL/kg/h.

#### 2.6.1. Phase 1

The response to TOF stimulation was assessed 100 s after the start of the infusion. If the TOF count was 4 (or any TOF ratio > 0%), the infusion rate was increased by 0.04 mL/kg/h, and an additional rapid IV bolus of 0.04 mL/kg was administered. This procedure was repeated every 2 min (approx. 100 s waiting time, 2 s for stimulation and evaluation, 18 s for bolus administration and infusion rate adjustment) until the TOF count decreased to ≤3 (i.e., no movement detected by the accelerometer in response to the fourth stimulation). The total cumulative dose of administered boluses (equal in number to the final infusion rate) was defined as the “induction amount” (mL/kg), marking the end of Phase 1.

#### 2.6.2. Phase 2

The NMBA infusion rate was then adjusted every 15 min following an up-and-down titration method according to TOF responses. If a TOF count of 4 (or any TOF ratio >0%) was observed, an IV bolus was administered with 20% of the induction amount (in mL/kg), and the infusion rate was increased by the same amount (in mL/kg/h). If a TOF count ≤3 was observed, the infusion rate was reduced by 10% of the induction amount, with no bolus administered. This continued until the end of the anesthetic event.

### 2.7. Statistical Analysis

The induction amount (mL/kg) and the further infusion rates (mL/kg/h) were recorded for all subjects. Body temperatures, ETCO_2_, and mean arterial pressures are presented as medians [interquartile range] of individual means for values collected during propofol administration. Medians [interquartile range] of coefficient of variations (CV%) are calculated for the mean arterial pressures (individual ratios of standard deviation over mean) to better illustrate its intra-individual variation. For each pig, a “final infusion rate” was calculated as follows. First, data collected during periods when ETCO_2_ was not maintained at the target value of 40 mmHg, as well as data from the subsequent 10 min, were excluded from further analysis. Next, a mean infusion rate was calculated between pairs of consecutive TOF stimulations whenever their results differed, resulting in a directional change in the infusion rate (i.e., a pair of TOF counts >3 and ≤3 resulting in a decrease followed by an increase in the infusion rate, or vice versa). Infusion rates that did not correspond to such directional changes were ignored. The individual final infusion rate was then determined as the average of the last four mean infusion rates.

For example, in a pilot pig, the induction amount (Phase 1) required to reach a TOF count ≤3 was 0.44 mL/kg/h (corresponding to six additional boluses). During Phase 2, the TOF response was re-evaluated every 15 min. After each evaluation, the infusion rate was either increased by 0.088 mL/kg/h (20% of the induction rate) with a bolus of the same amount for a TOF count >3 or decreased by 0.044 mL/kg/h (10% of the induction rate) for a TOF count ≤3. The anesthetic event lasted nearly 4 h, including 15 evaluations, with seven increases and eight decreases in the infusion rate. Of these, eight evaluations were excluded due to ETCO_2_ > 40 mmHg. Among the remaining seven, five pairs were identified that induced a directional change in the infusion rate. The mean infusion rate was calculated for each of these five pairs, and the average of the last four provided the final infusion rate (0.545 mL/kg/h).

For each animal, the cost of an IV bolus administered at the induction amount was calculated (CHF/kg). The individual unitary cost was the cost of an IV infusion at the final infusion rate (CHF/kg/h). The cost for 1 mL of rocuronium (10 mg/mL) was 1.040 CHF (0.104 CHF/mg). The cost for 1 mL of diluted atracurium (5 mg/mL) was 2.44 CHF (0.488 CHF/mg) without accounting for the cost of saline solution.

For each group, induction amount, final infusion rate, cost of the induction dose, and unitary cost are presented as median [interquartile range]. When judged by relevant amplitude, differences between groups for a specific variable were tested with a Wilcoxon signed-rank test. According to the recommendation from the American Statistical Association 13, *p*-values are reported without setting a cutoff for significance.

## 3. Results

All pigs were included in the study, leading to a total of six pigs per group, weighing 39.3 [31.3–46.4] and 28.7 [28.1–28.9] kg and aged 12 [11.1–12.9] and 10.5 [10.5–10.5] weeks in groups R and A, respectively (Table 1).

The stimulation intensity required to elicit an appropriate movement was between 35 and 50 mA in all pigs. The median predicted plasma concentration of propofol required for intubation was 22.0 [20.3–22.8] μg/mL and 19.7 [14.5–21.5] μg/mL in groups R and A, corresponding to a total administered dose of 21 [17–26] mg/kg and 22 [15–29] mg/kg, respectively. The median Prop_Target_DoA_0% was 8 [7–9] μg/mL and 7.25 [5–10] μg/mL in groups R and A, respectively. During Phase 2, body temperature was 38.9 [38.8–38.9] °C and 38.7 [38.4–38.9] °C, ETCO_2_ was 43 [41–44] mmHg and 44 [42–44] ± 2 mmHg, and mean arterial pressure was 90 [86–93] mmHg for 11 [11–13] CV%, and 73 [68–88] mmHg for 17 [15–18] CV% (*p* = 0.1797), for group R and A, respectively.

All pigs from group R required a single bolus of rocuronium (1 [1]), totaling 2 [2] mg/kg as an induction dose in Phase 1. In contrast, animals receiving atracurium required 8 [6–9] boluses to reach the induction dose of 2.3 [1.8–2.6] mg/kg. The price of an IV bolus at the induction dose was 0.208 [0.208–0.208] CHF/kg for group R, and 1.122 [0.878–1.366] CHF/kg for group A (*p* = 0.002725). In Phase 2, the final infusion rate of rocuronium administered was 4.5 [4.4–4.5] mg/kg/h, while that of atracurium was 2.7 [2.5–2.8] mg/kg/h (*p* = 0.004922). The unitary cost was 0.47 [0.45–0.47] CFH/kg/h in group R, compared to 1.30 [1.22–1.37] CFH/kg/h in group A (*p* = 0.0043).

## 4. Discussion

This study reports that the dose of atracurium needed to induce muscle relaxation is slightly higher than that of rocuronium, while the maintenance infusion rate was about 40% lower for atracurium than for rocuronium. Overall, administration of atracurium leads to significantly higher costs for both initial bolus and maintenance when targeting a similar muscle relaxation.

The present study reports that, in pigs, rocuronium may be markedly less expensive than atracurium to achieve appropriate muscle relaxation in Switzerland. Under the conditions presented here (the per-gram price of rocuronium being 21% that of atracurium), the cost of atracurium was 5.4 times higher than rocuronium for the initial bolus, and 2.8 times higher for maintenance infusion. This difference will vary with the purchase cost of the drugs (per gram). According to the dose equivalence obtained in the present study, atracurium would need to cost no more than 87% or 166% of rocuronium’s price (per gram) to be equally expensive during initial bolus and maintenance infusion, respectively.

The atracurium dose observed herein to reach deep muscular relaxation was 2.3 mg/kg for the initial dose, followed by 2.7 mg/kg/h. Previous information on the use of atracurium in pigs is scarce but seems in agreement with this result. Doses of 0.6 mg/kg or less were previously found inappropriate, and only a mild blockade was achieved [18]. According to Shorten et al. (1993) [12], a dose of 1.15 mg/kg allowed near-complete suppression of T1 (ED95% that was represented by 1150 ± 270 µg/kg of atracurium) in juvenile pigs (age range seven to nine weeks), indicating profound relaxation. In other publications, doses of 2 mg/kg and above were required to elicit deep neuromuscular blockade [19,20].

Regarding the use of rocuronium in pigs, an apparent result of the present study is that a similar or lower bolus dose is necessary compared to atracurium to induce neuromuscular relaxation, while a significantly higher dose is required for infusion maintenance. In previous studies, doses of rocuronium below 1 mg/kg were mostly judged inappropriate or insufficient [4,21,22,23]. Doses between 1 and 2 mg/kg proved sufficient to elicit moderate relaxation, including T1 depression of the TOF count for short durations [14,16,24,25]. Doses above 3 mg/kg induced prolonged deep muscle relaxation [26,27,28]. These observations are in agreement with the present study, where 2 mg/kg provided marked relaxation.

Regarding infusion rate, the current work reports that achieving optimal muscular blockade with a TOF count of 3 or lower required nearly double the maintenance amount (mg/kg/h) of rocuronium compared to atracurium. This is consistent with a previous study in juvenile animals where a dose of 2.5 mg/kg/h of rocuronium proved insufficient to elicit muscle relaxation, while 5 mg/kg/h was satisfactory [15]. The present study does not provide insights into the reasons for the difference in equipotent doses between the two drugs.

The dosages obtained in the present study are notably higher compared to those reported for other species for both atracurium [29] and rocuronium [30]. According to Clark-Price S., pigs appear to require significantly higher doses of NMBA than other domestic animal species [31]. However, evidence on this matter is lacking, and it remains unclear whether these differences result from pharmacokinetic or pharmacodynamic variations across species. Conversely, satisfactory neuromuscular relaxation was reached with pancuronium in pigs at doses comparable to those used in other species [32]. These observations highlight the need for dedicated, species-specific, evidence-based guidelines.

An interesting finding of the current analysis is the different number of boluses between groups required to achieve muscle relaxation. The objective was to find the appropriate dose to induce a deep muscle relaxation as targeted by a TOF count of 3. Therefore, a slightly lower dose than expected was administered for both drugs to allow for titrating it up without requiring too many boluses. However, the strategy was suboptimal for both drugs. For rocuronium, deep relaxation was achieved at the first dose of 2 mg/kg in all pigs, suggesting that a lower dose would have been sufficient. For atracurium, the first dose of 1 mg/kg was never sufficient and at least six additional boluses (0.2 mg/kg increments) were necessary in all pigs to reach the targeted effect, suggesting that a higher initial dose would have been more appropriate.

The high number of boluses prolonged the time between the first and the last administrations, potentially leading to a loss of effect from the initial doses. This suggests that a slightly lower total dose of atracurium may have been sufficient. For both groups, body temperature and end-tidal CO_2_ demonstrated stability within normal ranges during Phase 2. This is of importance as these variables (i.e., peri-anesthetic hypothermia or respiratory acidosis) have been reported to potentially influence the efficacy of NMBAs [33].

A significant difference was observed for the mean arterial pressure (MAP), with group R recording higher values (90 mmHg) compared to group A (73 mmHg). While more recently developed NMBAs are expected to exhibit mild to no cardiovascular effects, histamine release or sympathomimetic activity (i.e., affinity for cardiac muscarinic receptors) have been described and may vary across patients and situations [26]. However, the present study was not structured to answer this question, and no conclusion can be drawn.

Several limitations of this study should be acknowledged. First, the placement of the electrodes for the TOF assessment differed slightly compared to previous literature because it was adapted to experimental constraints. In prior porcine studies, ulnar nerve stimulation typically involves electrodes positioned on the palmar side of the forearm [12,15]. In the present study, however, the sling used to support the pig partially obstructed this configuration. As a pragmatic adaptation, electrodes were placed laterally, guided by visible muscle activation. While the amplitude of the neuromuscular responses may vary with electrode positioning relative to the targeted nerve, a clear and reproducible response was verified in each subject prior to data collection. All comparisons between groups were conducted under standardized conditions to minimize bias. Nevertheless, the impact of electrode placement variations on TOF results in pigs remains understudied and warrants further investigation.

A second limitation is the absence of pre-experimental calibration of the TOF-Watch^®^ device. Although manufacturer guidelines recommend calibration to account for individual variability in movement response gain, this step was omitted. To mitigate potential bias, according to guidelines published for reporting studies on the use of NMBAs [34], a standardized protocol was implemented: a fixed 2-Hz TOF stimulation for 1.5 s with a 12-s minimum latency, consistent current intensity, and unchanged stimulation duration throughout experiments. Stimulation intensity was initially adjusted to elicit a robust response in each animal (in particular to avoid excessive muscular response), and TOF counts were validated by visual observation during NMBA administration. Despite these measures, the lack of formal calibration may introduce unquantified variability.

Finally, the small sample size limits the generalizability of the findings. While the study provides preliminary insights into NMBA cost-effectiveness in swine, larger cohorts are needed to confirm these results and explore interindividual variability as well as the potential effect of demographics (age, sex, weights, etc.).

## 5. Conclusions

The results of this study demonstrate that rocuronium may offer a financial advantage over atracurium in inducing muscle relaxation in pigs. The final infusion rate of rocuronium (4.5 mg/kg/h) was higher than that of atracurium (2.7 mg/kg/h), but the unit cost for the infusion was significantly lower for rocuronium compared to atracurium. While cost differences may differ among places, the present study provides evidence for equipotent doses of the two NMBAs in swine.

## Figures and Tables

**Table 1 animals-15-01854-t001:** Main characteristics of the enrolled population.

Pigs Name	Group	Body Mass (kg)	Sex	Age (Weeks)
F	A	42.1	F	12
G	R	46.8	F	13
H	R	45.2	F	12.5
I	A	28	M	10.5
J	R	47.8	M	13.5
K	A	28.9	F	10.5
L	A	23.9	M	9
M	A	29	M	10.5
N	R	33.4	M	11.5
O	R	30.6	F	11
P	A	28.5	F	10.5
Q	R	28.9	F	10.5

Group: A: atracurium group; R: rocuronium group; Sex: M: male; F: female.

## Data Availability

Data is available on request.

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
