# Peer review of "Equipotent Dose and Cost Comparison of Atracurium and Rocuronium in Laboratory Pigs Anesthetized with Propofol"

_animals, 2025, doi:10.3390/ani15131854_

Round 1

Reviewer 1 Report

Comments and Suggestions for Authors

This report describes work comparing the cost effectiveness of two neuromuscular blocking agents administered to experimental pigs at doses and infusion rates producing comparable levels of neuromuscular blockade at the same nerve-muscle unit.

The methods used are sound and well-described.

The results are adequately detailed.

The discussion is good, although in the abstract the authors state, “ the study showed that pigs require higher doses than other species, highlighting the need for dedicated guidelines.”. This fact should be discussed to a greater extent than is currently present (Lines 254 – 259).

The work is original, useful and interesting and addresses a reasonably important question amongst a small number of researchers. It is a useful contribution to the field of NMB use in experimental pigs.

The conclusions are consistent with the evidence and arguments presented and address the main question posed, but need modification nevertheless; I would add that the study provides strong evidence for equipotent doses of the two NMB drugs, but that relative cost advantages depend on native drug costs. The current study revealed that rocuronium costs were significantly less when expressed in Swiss francs, but this may not be the case elsewhere.

All the references are appropriate.

In addition:

Line 17. For “starting dose” use “Loading dose”

Line 18: For cheaper use “Less expensive”

Line 80 NOT cross-over

Line 88.  Was the 1:1 sex ratio maintained in the two groups?

Line 268.  Let the reader decide if this is an interesting finding.

Comments on the Quality of English Language

Its a little clumsy in parts, but acceptable.

Author Response

The authors thank the reviewers for their work and suggestions. We hope this new version addresses these comments appropriately and provides an improved manuscript for the readership.

Answers to reviewers’ comments:

Reviewer 1:

  1. In the abstract the authors state, “ the study showed that pigs require higher doses than other species, highlighting the need for dedicated guidelines.”. This fact should be discussed to a greater extent than is currently present (Lines 254 – 259).
    Response: A short additional paragraph has been included in the discussion to raise this matter more specifically.
  2. I would add that the study provides strong evidence for equipotent doses of the two NMB drugs, but that relative cost advantages depend on native drug costs.
    Response: A sentence has been added in the conclusion.
  3. Line 17. For “starting dose” use “Loading dose”.
    Response: The authors believe that starting dose is appropriate for lay summary where readership may not be familiar with standard pharmacologic vocabulary.
  4. Line 18: For cheaper use “Less expensive”.
    Response: Changed.
  5. Line 80 NOT cross-over.
    Response: Cross-over removed (sorry for this mistake).
  6. Line 88.  Was the 1:1 sex ratio maintained in the two groups?
    Response: Not exactly. Inclusion F:M ratio was 7:5 (mistake corrected). It is mentioned in the M&Ms that sex was not considered for randomization, and the actual distribution is displayed in Table 1 (Group A: 3M, 3F; Group R: 2m, 4F). A short sentence has been added in the limitation part of the discussion (last sentence).
  7. Line 268.  Let the reader decide if this is an interesting finding.
    Response: The authors want to stimulate interest of the readership. A small change has been made to add precision to the statement.

Reviewer 2 Report

Comments and Suggestions for Authors
  • Ensure consistent use of terminology neuromuscular blocking agents/ non-depolarizing NMBAs.
  • 47-48 You state that the use of non-depolarizing NMBAs is commonplace in surgical procedures in pigs—implying a routine or standard practice in biomedical research—yet later indicate they were administered in only approximately 20% of cases [6]. This discrepancy should be clarified.
  • A possible additional bibliographical source to support the research is:

Veres-Nyéki KO, Rieben R, Spadavecchia C, Bergadano A. Pancuronium dose refinement in experimental pigs used in cardiovascular research. Vet Anaesth Analg. 2012 Sep;39(5):529-32. doi: 10.1111/j.1467-2995.2012.00732.x. Epub 2012 Apr 4. PMID: 22486886.

Author Response

The authors thank the reviewers for their work and suggestions. We hope this new version addresses these comments appropriately and provides an improved manuscript for the readership.

Answers to reviewers’ comments:

Reviewer 2:

  1. Ensure consistent use of terminology neuromuscular blocking agents/ non-depolarizing NMBAs.
    Response: Consistent use has been checked and homogenized.
  2. Line 47-48: You state that the use of non-depolarizing NMBAs is commonplace in surgical procedures in pigs—implying a routine or standard practice in biomedical research—yet later indicate they were administered in only approximately 20% of cases [6]. This discrepancy should be clarified.
    Response: “Commonplace” has been changed for “common” to better fit to 20%, and a comparison to veterinary practice added.
  3. A possible additional bibliographical source to support the research is: Veres-Nyéki KO, Rieben R, Spadavecchia C, Bergadano A. Pancuronium dose refinement in experimental pigs used in cardiovascular research. Vet Anaesth Analg. 2012 Sep;39(5):529-32. doi: 10.1111/j.1467-2995.2012.00732.x. Epub 2012 Apr 4. PMID: 22486886.
    Response: Thanks, added when discussing dose studies.

Reviewer 3 Report

Comments and Suggestions for Authors

General comments

Anesthetic management in large species is becoming increasingly important due to the lack of certain information on dosage, kinetics and, above all, clinical efficacy from an experimental or clinical point of view. Therefore, this article contributes actively in this area, and for this reason I consider this proposal to be innovative and appropriate for this area of study. However, a significant weakness of this proposal is the lack of adequacy of the text according to the guidelines of the journal, therefore I suggest that they should be corrected.

Response:

Another weakness of this proposal could be to establish the relationship of cost to benefit, I suggest that this approach could increase the impact of your proposal.

Response:

Finally, another significant weakness is the lack of clarification of the experimental design used in your study because you do not state the inclusion and exclusion criteria for the animals assigned in the study groups. This would help to better understand your study.

Response:

Particular comments

Line 2. I agree with your proposed article title, however, if the authors allow me I suggest that it could be modified to “evaluation of the cost and equipotent dose of two of atracurium and rocuronium in healthy pigs anesthetized with propofol under experimentation”.

Response:

Line 13. Please can you clarify if these were healthy animals?

Response:

Line 14. Was the distribution of the animals done randomly? Please clarify.

Response:

Lines 16- 17. This statement is confusing, if the authors allow me I suggest you clarify if the requirement of atracurium was significantly lower compared to rocuronium, but, it was also noted that the cost was significantly higher for atracurium compared to rocuronium. Perhaps this sentence can be clearer for your simple summary.

Response:

Line 23. Please replace the “;” with a “.”

Response:

Lines 25- 27. These sentences are confusing and clarify the number of animals destined in each group and the number of groups if the authors allow me I suggest modifying it as “equipotent dose for atracurium (A) and rocuronium (R) in laboratory pigs. 12 healthy laboratory pigs were randomly distributed into two study groups according to the muscle relaxant administered: group A: atracurium (n= 6) and group B: rocuronium (n= 6)”. This would possibly clarify in a general way the design of their study.

Response:

Line 29. Please eliminate the space between paragraphs. An abstract should be a single paragraph.

Response:

Line 30. Replace “Initial induction” with “bolus of induction”, it is the more conventional term.

Response:

Line 40. I agree with your keywords, however, if the authors allow me I suggest that you add the term “pharmacoeconomics” This could increase your chances of matching in the different databases as your study has a strong relation with this area.

Response:

Line 41. As a general recommendation, I suggest that authors take care and respect the suggested format of the journal with respect to the alignment of the text. Also, as another recommendation, I suggest that you divide this section into 3 or more paragraphs because a single paragraph can be tiring for the reader. Please consider this recommendation.

Response:

Lines 44- 47. This statement is very interesting, however, it is not clear what is the advantage of the use of muscle relaxants in this species. If the authors allow me I suggest that they mention examples of surgical procedures where relaxants facilitate this type of procedure such as ophthalmic surgery.

Response:

Lines 47- 50. These lines serve to make known the types of muscle relaxants, therefore, I suggest that you mention what type of relaxants and the general mechanism of action of these that complement those described in the first paragraph.

Response:

Line 54. Please add the references of the mentioned studies.

Response:

Lines 57- 63. Is what is described in these lines described in pigs or other species? Because it is not clear if the tone in these lines refers to the lack of information in swine, please clarify this justification of your study.

Response:

Line 67 Please try to tailor your objective as it differs from what is described above.

Response:

Line 69. If the authors allow me I suggest that you add a hypothesis that can complement your objective.

Response:

Line 72. Please I suggest that this information be included in a section called ethical statement.

Response:

Lines 76- 79. I do not find it necessary to mention this information, if the authors agree I suggest it be removed.

Response:

Lines 79- 83. As a complement to my previous comment, I suggest that this information be included in a section called experimental design where they mention the groups assigned, if the distribution was randomized and the variables evaluated.

Response:

Line 87. Could you clarify if the animals admitted to the study were healthy? In complement to this comment, I suggest that you describe what were the criteria for inclusion and exclusion of the animals.

Response:

Line 93. What type of examination was performed? General physical examination? Did you perform hematologic studies? Did these studies help establish the anesthetic ASA risk of the animals? Please suggest that this information should be clarified.

Response:

Line 96. If the authors allow me I suggest that this information be included in the subsequent section to form a section called anesthetic and anesthetic management.

Response:

Line 101. This information is incomplete, please suggest you mention if the catheterization was aseptic, what type of catheter was used, was this catheter used to administer fluids, what type of fluid was, what infusion rate was used, and did you use an infusion pump.

Response:

Line 103. Again this information is incomplete. Was premedication of the animals performed? Did they apply any tranquilizers? Was the anesthetic induction with propofol performed? If so, what dose was used? And what was the continuous infusion dose used? I realize that this information is possibly off-target, but, I invite the authors to provide accurate animal handling information and have this information accompanied by references.

Response:

Line 109. Again what were the ventilation parameters used? What type of machine did you use? Was the fan integrated into the machine or was it independent? If so please clarify the model of the fan.

Response:

Line 112. Add reference.

Response:

Line 113- 114. This statement is not clear, did you use a pharmacometric predictive method for propofol? If so please suggest that they clarify this methodology, in addition to mentioning that the assessment of anesthetic depth was assessed by evaluating clinical reflex signs.

Response:

Line 126 Please use this information to resolve the doubt mentioned in previous comments.

Response:

Line 127. Was anesthetic management performed by the same anesthetist throughout the study?

Response:

Line 210. What was the significance level you established in your analysis?

Response:

Lines 216- 231. The description of the results mentioned in this section differs from those mentioned in your summary. I suggest that you verify that all the information is mentioned and that it is clear whether these differences are significant. On the other hand, it is important that the tables are mentioned or cited in the text. In this sense, if the authors agree I suggest that to give more understanding to their results it would be convenient to include a table that helps to show the results of their variables.

Response:

Line 235. This statement is interesting, you failed to mention what was observed with costs Could you discuss or provide a biological explanation as to why there was a lower requirement for atracurium? Is it related to the pharmacokinetic characteristics of these drugs? Please discuss.

Response:

Lines 238- 289. Again a paragraph too wide, please, I invite the authors to generate smaller paragraphs dividing the information. This may make it easier to read.

Response:

Author Response

The authors thank the reviewers for their work and suggestions. We hope this new version addresses these comments appropriately and provides an improved manuscript for the readership.

Answers to reviewers’ comments:

Reviewer 3:

  1. A significant weakness of this proposal is the lack of adequacy of the text according to the guidelines of the journal, therefore I suggest that they should be corrected.
    Response: Alignment has been corrected, and the paragraphs have been shortened / separated to ease reading.
  2. Another weakness of this proposal could be to establish the relationship of cost to benefit, I suggest that this approach could increase the impact of your proposal.
    Response: The authors mention in the introduction that there is little pharmacological benefit associated with the choice of NMBAs in standard situations, therefore the cost may play a relevant role.
  3. You do not state the inclusion and exclusion criteria for the animals assigned in the study groups. This would help to better understand your study.
    Response: Inclusion and exclusion criteria have been added.
  4. Line 2. I agree with your proposed article title, however, if the authors allow me I suggest that it could be modified to “evaluation of the cost and equipotent dose of two of atracurium and rocuronium in healthy pigs anesthetized with propofol under experimentation”.
    Response: The title has been changed.
  5. Line 13. Please can you clarify if these were healthy animals?
    Response: Added.
  6. Line 14. Was the distribution of the animals done randomly? Please clarify.
    Response: Details are available in the M&Ms.
  7. Lines 16- 17. This statement is confusing, if the authors allow me I suggest you clarify if the requirement of atracurium was significantly lower compared to rocuronium, but, it was also noted that the cost was significantly higher for atracurium compared to rocuronium. Perhaps this sentence can be clearer for your simple summary.
    Response: The sentences specify clearly that the dose was lower but the cost higher for atracurium. A small change has been done to reduce confusion.
  8. Line 23. Please replace the “;” with a “.”
    Response: Changed.
  9. Lines 25- 27. These sentences are confusing and clarify the number of animals destined in each group and the number of groups if the authors allow me I suggest modifying it as “equipotent dose for atracurium (A) and rocuronium (R) in laboratory pigs. 12 healthy laboratory pigs were randomly distributed into two study groups according to the muscle relaxant administered: group A: atracurium (n= 6) and group B: rocuronium (n= 6)”. This would possibly clarify in a general way the design of their study.
    Response: Changed.
  10. Line 29. Please eliminate the space between paragraphs. An abstract should be a single paragraph.
    Response: Changed.
  11. Line 30. Replace “Initial induction” with “bolus of induction”, it is the more conventional term.
    Response: Changed.
  12. Line 40. I agree with your keywords, however, if the authors allow me I suggest that you add the term “pharmacoeconomics” This could increase your chances of matching in the different databases as your study has a strong relation with this area.
    Response: Added.
  13. Line 41. As a general recommendation, I suggest that authors take care and respect the suggested format of the journal with respect to the alignment of the text. Also, as another recommendation, I suggest that you divide this section into 3 or more paragraphs because a single paragraph can be tiring for the reader. Please consider this recommendation.
    Response: Changed.
  14. Lines 44- 47. This statement is very interesting, however, it is not clear what is the advantage of the use of muscle relaxants in this species. If the authors allow me I suggest that they mention examples of surgical procedures where relaxants facilitate this type of procedure such as ophthalmic surgery.
    Response: Added, but this remains low evidence and ophthalmic surgery is rare in pigs.
  15. Lines 47- 50. These lines serve to make known the types of muscle relaxants, therefore, I suggest that you mention what type of relaxants and the general mechanism of action of these that complement those described in the first paragraph.
    Response: A sentence on mechanism is added in the introduction.
  16. Line 54. Please add the references of the mentioned studies.
    Response: The references are given in the following sentences.
  17. Lines 57- 63. Is what is described in these lines described in pigs or other species? Because it is not clear if the tone in these lines refers to the lack of information in swine, please clarify this justification of your study.
    Response: The sentences clearly refer to “use in this species” and “porcine medical research”.
  18. Line 67 Please try to tailor your objective as it differs from what is described above.
    Response: The objective is stated to compare dose and cost between the two drugs.
  19. Line 69. If the authors allow me I suggest that you add a hypothesis that can complement your objective.
    Response: An hypothesis has been added.
  20. Line 72. Please I suggest that this information be included in a section called ethical statement.
    Response: Added.
  21. Lines 76- 79. I do not find it necessary to mention this information, if the authors agree I suggest it be removed.
    Response: The authors believe this is a relevant information for transparency.
  22. Lines 79- 83. As a complement to my previous comment, I suggest that this information be included in a section called experimental design where they mention the groups assigned, if the distribution was randomized and the variables evaluated.
    Response: Added.
  23. Line 87. Could you clarify if the animals admitted to the study were healthy? In complement to this comment, I suggest that you describe what were the criteria for inclusion and exclusion of the animals.
    Response: Added.
  24. Line 93. What type of examination was performed? General physical examination? Did you perform hematologic studies? Did these studies help establish the anesthetic ASA risk of the animals? Please suggest that this information should be clarified.
    Response: Some details added. No other examination (e.g. hematologic) was performed than these mentioned.
  25. Line 96. If the authors allow me I suggest that this information be included in the subsequent section to form a section called anesthetic and anesthetic management.
    Response: Not changed.
  26. Line 101. This information is incomplete, please suggest you mention if the catheterization was aseptic, what type of catheter was used, was this catheter used to administer fluids, what type of fluid was, what infusion rate was used, and did you use an infusion pump.
    Response: Details added (or transferred between paragraphs).
  27. Line 103. Again this information is incomplete. Was premedication of the animals performed? Did they apply any tranquilizers? Was the anesthetic induction with propofol performed? If so, what dose was used? And what was the continuous infusion dose used? I realize that this information is possibly off-target, but, I invite the authors to provide accurate animal handling information and have this information accompanied by references.
    Response: Details added in methods and in results sections.
  28. Line 109. Again what were the ventilation parameters used? What type of machine did you use? Was the fan integrated into the machine or was it independent? If so please clarify the model of the fan.
    Response: Details added.
  29. Line 112. Add reference.
    Response: The authors are not sure to understand which statement requires a reference here.
  30. Line 113- 114. This statement is not clear, did you use a pharmacometric predictive method for propofol? If so please suggest that they clarify this methodology, in addition to mentioning that the assessment of anesthetic depth was assessed by evaluating clinical reflex signs.
    Response: Evaluated signs of anaesthetic depth are mentioned. Details on the TCI are added.
  31. Line 126 Please use this information to resolve the doubt mentioned in previous comments.
    Response: Moved
  32. Line 127. Was anesthetic management performed by the same anesthetist throughout the study?
    Response: Detail added.
  33. Line 210. What was the significance level you established in your analysis?
    Response: As mentioned in the article, according to the recommendation from the American Statistical Association, p values are reported without setting a cutoff for significance.
  34. Lines 216- 231. The description of the results mentioned in this section differs from those mentioned in your summary. I suggest that you verify that all the information is mentioned and that it is clear whether these differences are significant. On the other hand, it is important that the tables are mentioned or cited in the text. In this sense, if the authors agree I suggest that to give more understanding to their results it would be convenient to include a table that helps to show the results of their variables.
    Response: The authors apologize, there was a mistake here. A sentence of the results was erroneously deleted. The result section is complete now.
  35. Line 235. This statement is interesting, you failed to mention what was observed with costs Could you discuss or provide a biological explanation as to why there was a lower requirement for atracurium? Is it related to the pharmacokinetic characteristics of these drugs? Please discuss.
    Response: The doses observed in the present study are discussed lower in the discussion part. However, it remains impossible for the authors to elucidate if the reasons are PK- or PD-related or influenced by other bias, as this was not investigated here. The discussion part on this topic has been precised.
  36. Lines 238- 289. Again a paragraph too wide, please, I invite the authors to generate smaller paragraphs dividing the information. This may make it easier to read.
    Response: Paragraphs have been divided.

Round 2

Reviewer 3 Report

Comments and Suggestions for Authors

General comments

I appreciate that the authors have considered my comments on their proposed manuscript, which undoubtedly contributes to the field of development. However, there are still minimal adjustments to be made for its improvement.

Response:

Particular comments

Line 82. Please, I suggest that you unify the method for referring to the dosage, as you used mg/kg-1 here but later used mg/kg. I suggest that you decide which one you will use.

Response:

Line 105. Please specify that it was ASA 1 or anesthetic risk 1, not just 1.

Response:

Final comments

I suggest that in your methodology you review the suggested journal format because you recommend using subtopic numbering to give order to each one.

Response:

Author Response

Sorry for these last changes required. These have been performed.

Line 82. Please, I suggest that you unify the method for referring to the dosage, as you used mg/kg-1 here but later used mg/kg. I suggest that you decide which one you will use.

Response: mg.kg-1 has been changed for mg/kg

Line 105. Please specify that it was ASA 1 or anesthetic risk 1, not just 1.

Response: done

Final comments

I suggest that in your methodology you review the suggested journal format because you recommend using subtopic numbering to give order to each one.

Response: done.